# Understanding the Human Dimensions of Recycling and Source Separation Practices at the Household Level: An Evidence in Perak, Malaysia

**Pei Lin Yu [1], Norafida Ab Ghafar [2], Mastura Adam [2] and Hong Ching Goh [1,*]**

[1] Department of Urban and Regional Planning, Faculty of Built Environment, Universiti Malaya, Kuala Lumpur 50603, Malaysia; yu_peilin@hotmail.com
[2] Department of Architecture, Faculty of Built Environment, Universiti Malaya, Kuala Lumpur 50603, Malaysia; norafida@um.edu.my (N.A.G.); mastura@um.edu.my (M.A.)
* Correspondence: gohhc@um.edu.my

**Abstract:** Recycling and source separation (R&SS) are believed to have been the first attempt to minimise waste. This research adopted mixed methods that followed sequential quantitative then qualitative data collection, combining questionnaire surveys from 100 households, semi-structured interviews, and participatory observations to study the human dimension of waste generation and management. Scoring Assessment (with modified Bloom's Cut Off point) indicated that households had moderate knowledge and positive attitudes yet poor behaviour, and these three components indicated no linear associations, tested using Pearson's Correlation Coefficient. However, age group, marital status, educational level and living duration showed statistical significance with households' participation in source separation through Chi-Square Test. Meanwhile, observation data showed that waste management mechanisms and environment had inefficiently supported households' participation in R&SS practices (external factors: poor accessibility to services, lack of tangible incentives, and absence of restriction in consumption). Elicited data indicated that a satisfactory level of intentions, knowledge, and willingness, together with good habit and quality persuasion (internal factors), were required to drive good behaviour. Subsequently, a series of recommendations were formulated to promote gradual yet solid transformation of the waste management system, tapping on existing initiatives by considering additional parameters upon the gap in households' knowledge, attitude, and behaviour.

**Keywords:** mixed methods; online questionnaire survey; participatory observation; sustainable waste disposal; sustainable consumption; Malaysian source separation practice

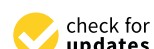

## 1. Introduction

"The throw-away society is a human society strongly influenced by consumerism. The term describes a critical view of overconsumption and excessive production of short-lived or disposable items", quoted [1], who argues the cost of this throw-away culture in compromising the needs of future generations and threatening the natural system that the survival of all living things depends on [2]. As more products are made more affordable, they are less appreciated, as society can dispose and buy new items, often beyond what is needed, rather than send them for repair [3]. This marks the peak of the global waste generation at 2.01 billion tonnes (0.74 kg per person daily), but its rate, amount, and quality will continue to surge by 70%, without consideration of the concept of distancing in dealing with waste during post- and pre-consumerism [4,5].

Statistics show that over 90% of waste in low-income countries (compared to 66% for low-middle-income and 30% for upper-middle-income) is disposed of at open dumps or landfills, which are the most adopted waste disposal methods [5,6]. These waste disposal sites have thus become the only and popular method used by cities (with limited municipal

budget) to dispose of the high volume of (unsorted) waste generated daily. Waste disposal sites are optimal breeding grounds for disease vectors and sources of toxin release into the atmosphere and oceans [2,5]. However, they will not be publicly acknowledged as environmental issues if landfilling remains the only urban waste management service [7]. If waste continues to be collected regularly without a proper sorting system to support it, and if there are no restrictions on consumption or changes in lifestyle choices, society will continue to remain in denial and ignorant to the over exploitation and destruction of the planet's natural systems as a result of their personal waste footprint [1,2].

Malaysia is one of the upper-middle-income countries that heavily rely on landfill disposal, with almost 89% of waste collected (from a waste generation rate at 33,130 tonnes daily) being sent to a total of 170 landfills. Out of these, only 14 are categorised as sanitary. The official lab report of the country estimates that at least 40% waste diversion can be achieved [8,9]. Consequently, space and land availability will gradually emerge as a major limitation to landfilling as the increasing waste volume exceeds the capacity of the treatment, not to mention other waste problems to be addressed, such as illegal dumping and plastic waste import [10,11]. The overconsumption of the throw-away society, together with almost-absent resource recovery attempts and a lack of political will and social responsibility towards sustainable and integrated waste management, present a huge barrier for the transition to waste minimisation [7].

The first effective step towards waste minimisation in the European Union waste hierarchy is recycling. This includes composting, which deals with more waste fractions, for instance, organic or biodegradable waste and e-waste [12,13]. The integration of recycling and source separation (R&SS) is crucial to create a compound effect on the waste diversion from landfilling when waste composition and its quality is carefully managed [14]. However, Malaysia's recycling efforts and implementation have focused on only a few categories of recyclables, while source separation only came into enforcement in late in 2015, along with the formulation of Solid Waste and Public Cleansing Management Act 2007 (Act 672) [7]. A community survey showed that only 28% of households in Kuala Lumpur engage in source separation, even though the legislation has been enforced [15]. This result denotes several constraints found in the implementation, especially when dealing with the complexity of human dimensions within the current waste management system and mechanism [7].

This contrasts with a case reported in Shanghai, China, where source separation was mandated in 2019. The study in [16] reports that nearly half of the households had negative emotions towards the policy. Although the households found it difficult to follow the segregation guidelines, which were rather broad and ambiguous in the details [17], it was fear towards the heavy fines imposed by the government rather than environmental protection or sustainability that pressured the households to comply with the regulation [16]. Thus, it deviated from the Chinese government's initial vision of promoting sustainable development, which aims to depreciate the culture of throwaway consumerism [18]. The key idea here is to promote a sustainable society by sustainable waste disposal while looking into individual daily consumption through lifestyle choice [1,3,18]. However, diversified urban governances and management systems, as well as different cultural readiness, lead to different speed and quality of transition to waste minimisation [17,19].

As such, it is important to investigate the potential factors that drive households' behaviour change and their potential adaptation to local context. In identifying these factors, many researchers relate the discussion of waste management and sustainable consumption with pro-environmental behaviour; the studies in [20,21] suggest that environmental knowledge is important yet insufficient to drive action. The studies in [22,23] argue that a high satisfactory level of knowledge, together with attitude, could more likely drive good behaviour, while [24] explains behaviour is an interactive output of attitude and choice with the presence of external causality such as constraint (cost, time), habits or routine, disincentives and scepticism. The study in [20] also claims that pro-environmental behaviour involves both internal and external factors. Meanwhile, [25] categorises non-recyclers into

three major groups based on the common characteristics of the barriers (or "excuses", as referred in the original article) selected through a community survey conducted by Ipsos in 2011. Each group discusses distinctive reasons and psychology behind the action towards recycling participation; for instance, time consumption, issue of convenience, lack of knowledge, or absence of communal effort or strong influence. These parameters guided them to weigh their decision together with the current waste management environment they are provided with. The interpretation of each grouping, as well as the interventions the author proposed, conveys the relation of multiple internal and external factors (potential parameters other than knowledge and attitude level) with the behaviour.

This study aims to make locally adaptive recommendations to encourage household participation in R&SS practices with a case study in Manjung district, Perak. In doing so, the study examined household KAB, as well as other potential parameters that influence household behaviour towards waste generation and management. Specifically, authors ask (1) What are the associations between the households' level of KAB towards R&SS practices? (2) What are the enabling factors of households' participation in R&SS practices? (3) What are the barriers of non-recyclers or non-waste sorters? (4) What are the practical recommendations to encourage R&SS participation?

This study contributes to empirical evidence by addressing the gap between the households' awareness and the actual sustainable waste disposal rate that relates directly with the households' participation in R&SS practices. The authors advocate the idea that changes in behaviour start with improved level of knowledge and attitude, although improved level of knowledge and attitude may not necessarily lead directly to change in behaviour. At a minimum, the public will have some knowledge about the generation (which would influence the purchasing decision and the material leftover after consuming products) and disposal of domestic waste so that the demand for products will be shifted to those that carry sustainable and environmentally friendly qualities and are easier to manage during disposal. This would have an impact on the supply side of materials, of which the manufacturers may respond to the demand and redesign the products using eco-friendly material. Consequently, when the entire chain of production adopts the concept of sustainability from such market activity, the system could then be elevated to a higher level in the waste hierarchy: reduce, reuse, and lastly, prevention.

By identifying households' knowledge level, their attitude towards both practices and the waste management process, as well as their behavioural pattern on the R&SS practices, this research intends to establish a basis to support the decision making process in relation to the waste management system. The proposed recommendations highlight the importance of further encouraging household involvement on R&SS practices. Discussions of results concerning expectation and feedback are useful for the municipality and relevant stakeholders for their service improvement and to help policymakers, waste management planners, local administrators, and researchers to formulate policies and strategies in sustainable waste management, as well as serve as a basis to identify further areas of study.

## 2. Materials and Methods

### 2.1. Case Study

Sanitation (and therefore waste management) is a matter under the Concurrent List. The state government has the authority to decide whether to adopt the law related to this urban management service, thus it is subject to the administration of each local authority. However, regarding the current governance status in Malaysia, the decision of mandating the Solid Waste and Public Cleansing Management Act 2007 (Act 672)—the latest source separation initiative—may vary over a short period of time. Perak is one of eight states (i.e., Penang, Perak, Terengganu, Kelantan, Selangor, Sabah, Sarawak, and Labuan) that are yet to mandate the legislative provision of Act 672 [26]. Without law enforcement to reduce the waste sent to landfills at state level, the capacity will eventually run out, with Perak's 0.8842 kg waste generation per capita per day (2002) generated by its 2.30 million population. This is a higher rate compared to Selangor at 0.8845 kg, yet Selangor has almost

double the population of Perak. At the local level, Manjung (Figure 1) has exceeded the national average (0.8500 kg per capita per day) at 1.409 kg, and ranks as the third highest among the districts [27].

Only a 4.00% recycling rate has been recorded by the study in [28] through the provision of a recycling service in the Northern Region (Kedah, Perlis, and Perak). On the other hand, the recorded recycling rate from the database of Manjung Municipal Council is 0.07%, based on the quotient of the total recyclables collected: approximately 77 tonnes per annum from the total waste collected from 54,186 housing, which is about 300 tonnes per day [29]. Both the state (region) and local rate of recycling are still distant from that of the national target: 22.00% in 2020. Being the third most populated city in Perak, Manjung has yet to generate sufficient awareness of the waste crisis and sustainable waste management [30].

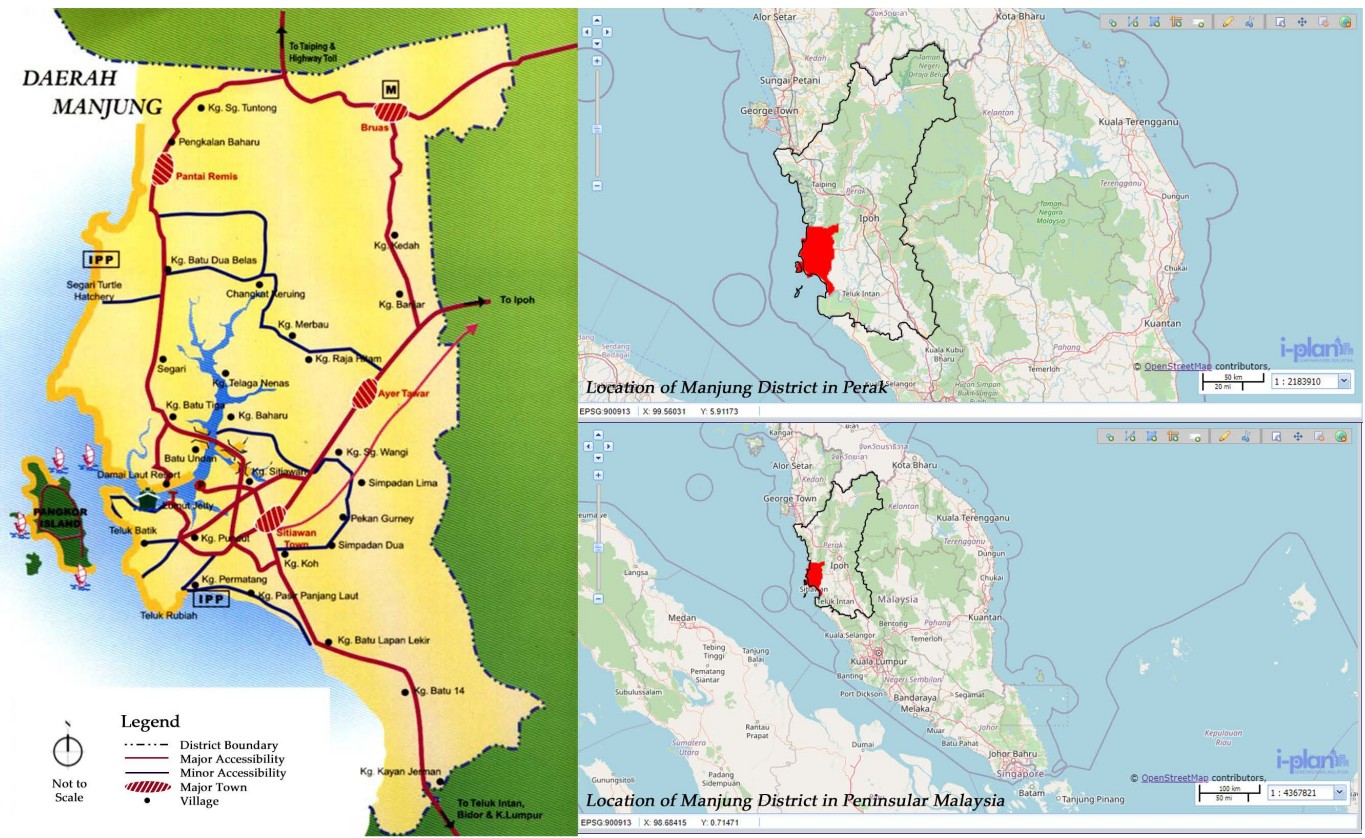

**Figure 1.** Location of Manjung District [31,32].

## 2.2. Data Collection and Sampling

This study adopted a mixed-method approach, combining an online questionnaire survey, semi-structured interviews (through meeting application or phone call) and participatory observation, adjusting to pandemic situations. The purposive sampling strategy was used throughout the data collection process (except for observation).

Before the actual conduct, pre-testing was carried out to examine the content validity of the questions asked in the survey (including semi-structured interview). All questions were refined based on the feedback from both experts in the field and laymen and were adapted from studies in past literature [30,33–39] to improve their reliability and representativeness to the study. The questionnaire survey was constructed using an online survey administration application, Google Form, and distributed through various online platforms (e.g., Facebook, Instagram, and WhatsApp, popular applications in Malaysia). The questions were written in three major languages: Malay, Chinese, and English. Respondents were filtered carefully under two conditions: a person (1) aged 18 years old and above who (2) has lived in Manjung district for more than a week. It was also mentioned that each

respondent would be representing a household. The questionnaire consisted of six main sections: (a) respondent's profile, (b) knowledge, (c) attitude, (d) behaviour, (e) psychology behind R&SS practices, and (f) invitation to interview. The targeted sample size for the survey was 382 respondents for 58,186 house premises (data from Manjung Municipal Council) based on the calculation by [40]; however, it was not achieved due, in part, to the relatively less effective channel of survey distribution via online networks. Only 100 respondents took part in the survey within the scheduled time frame for quantitative data collection. The mixed methods of this study underwent a sequence arrangement [41], in which the qualitative data that followed were collected after quantitative data were completed.

Seven respondents, who had answered the survey, were recruited to a semi-structured interview based on their consent and willingness to contribute to this study. This further strengthened the perspective of service users through descriptive and abstract primary data on their waste management process [42]. Questions to them were focused on the advantages and disadvantages of the R&SS, regardless of their actual participation. On the other hand, the semi-structured interview included the perspectives of the service providers to explain the condition and quality of R&SS practices provided, as well as how the households' needs were addressed. Two personnel from the local authority (the head of recycling project implementation) and non-governmental organisations (which are actively involved in R&SS implementation at household level) were interviewed. Questions for service providers merely focused on the mechanisms involved in providing their respective R&SS services and how to engage with the community.

Observation was conducted to inspect the current waste management mechanism and environment of the study area, from waste generation, storage, collection to disposal. Taking into consideration that presence of the observers might have affected the behaviour of the observed group, participatory observation was opted to blend in the situation [43]. Observation took place in public areas, including wet markets and neighbourhood streets, when municipal waste collection was in action. The observed objective focused on how humans carry out waste disposal, rather than on the humans themselves. Therefore, no consent was needed, and no personal communication was conducted. No photos were taken of any individual present at the observation spot.

### 2.3. Data Analysis

Since the data collected consisted of both quantitative and qualitative data, appropriate analysis techniques were used for each type of data accordingly.

Microsoft Excel 2019 was used to perform all statistical (quantitative) analysis, ranging from descriptive, inferential statistics to correlational analysis. Measures of central tendency and dispersion (mean and standard deviation) were used to study data distribution. In addition, the Chi-Square test, together with Pearson's Correlation Coefficient, which was performed at 95% confidence level, were used to determine the association between variables (two forms: nominal and continuous data). A scoring system was developed to assess the adequacy level of the KAB through a fixed range of scores modified according to the widely adopted Bloom's Cut-Off Point [38,44]. This method allowed the authors to convert different types of data, including nominal—true or false answer and percentages, and ordinal data—Likert scales, into scores—continuous data. The scores (by level) obtained were used to test the correlation between KAB components and other variables.

For knowledge (first component), 20 questions were asked in two parts: ten true or false questions (general knowledge assessment) and ten multiple choice questions (specific knowledge assessment). A score of 1 was given to each correct answer (for each sample); in contrast, no score was deducted for a wrong answer, instead it was given a score of 0. Hence, each part had a maximum and minimum score of 10 and 0, respectively. These scores were divided into three levels through a modified Bloom's Cut-Off Point, namely (a) High for 10 to 8 scores, (b) Moderate for 7 to 5 scores, and (c) Low for scores less than 5. For attitude (second component), there were 14 questions with a 5-Point Likert scale to assess the degree of importance and agreement. Scores 1 to 5 were given to respective points

on the Likert scale (in the order from Strongly Disagree to Strongly Agree). This resulted in a maximum score of 70 and a minimum of 14 for each sample. Modified Bloom's Cut-Off Point was also used to establish three levels, namely (a) Positive (70 to 52), (b) Neutral (51 to 33), and (c) Negative (14 to 32), which were equally divided and given the same class interval. Finally, for behaviour (third component), there were 10 questions with multiple choice of answers. Similar to the first component, the scores were divided into three levels through a modified Bloom's Cut Off Point, namely (a) Good (10 to 8), (b) Satisfactory (7 to 5), and (c) Poor (4 to 0).

Phenomenological (qualitative) analysis was used to carefully record the conduct of waste management by the service users during the observation and semi-structured interview processes, respectively. This was to ensure that both observed and elicited data could be precisely analysed and transformed into useful information for the description of a phenomenon. Additionally, it could extract the perception of the service providers on the outcomes of the waste management service [45].

### 2.4. Research Constraints and Limitations

First, respondents could not be observed while answering the survey during online pre-testing. This led to limited sources of input to improve the questionnaire; therefore, a more comprehensive manner of conducting pre-testing in both online and offline platforms is recommended to extract both observed and verbal feedback. The difficulty in recruiting sufficient samples during the lockdown due to the pandemic in Malaysia was the key constraint faced by the authors. Nevertheless, this should serve as a preliminary study, and similar studies covering a comprehensive sample size targeting 382 respondents are recommended post-pandemic. Second, the number of interviewees recruited from the service providers was limited due, in part, to the lack of effective ways to engage, as no physical visit was allowed due to movement restrictions. No positive response was obtained from private recycling vendors, probably due to the lack of awareness and exposure towards the academic research in the related field. Finally, invitation (survey or interview) through online networks was likely neglected, but this method still had benefits such as receiving responses quicker and in a more convenient way, therefore a combination of online and on-site survey distribution and interview invitation were proposed.

## 3. Results

### 3.1. Demographic Profile

A total of 100 questionnaire surveys were eligible to be analysed. Most of the respondents are between 18 and 24 years old with pre-university or undergraduate academic qualifications, female, and single. More than half of the respondents are locals who have lived in the community for more than 20 years. Nearly all the respondents live with their family members on landed properties, with the majority having a household size between 5 and 7 persons (Table 1).

### 3.2. Knowledge Assessment

Both general and specific knowledge levels were tested among the respondents. In general, the knowledge level among the respondents was rated as moderate (Table 2) according to the scores obtained from Tables 3 and 4, with better scores in general knowledge, where 37.0% scored high.

Table 3 shows the general knowledge among respondents of the R&SS with their benefits and importance to the waste management system, quality environment, and energy consumption. Most of the respondents answered correctly, except for statement #9: all plastics that contain numbered symbols (also known as plastic resin identification codes) can be recycled.

**Table 1.** Demographic profile of the respondents.

| Variables (*n* = 100) | % | Variables (*n* = 100) | % |
|---|---|---|---|
| Gender | | Duration of Stay | |
| Female | 73.0 | Less than a year | 4.0 |
| Male | 27.0 | 1–5 years | 9.0 |
| Marital Status | | 6–10 years | 11.0 |
| Single | 87.0 | 11–15 years | 7.0 |
| Married | 13.0 | 16–20 years | 17.0 |
| Divorced/Widowed | 0.0 | More than 20 years | 52.0 |
| Age Group | | Household Size | |
| 18 to 24 years old | 71.0 | 1 | 3.0 |
| 25 to 44 years old | 26.0 | 2–4 | 39.0 |
| 45 to 64 years old | 3.0 | 5–7 | 56.0 |
| 65 years old and above | 0.0 | More than 7 | 2.0 |
| Educational Level | | Housing Type | |
| Primary Education | 0.0 | Bungalow/Semi-Detached | 50.0 |
| Secondary Education | 4.0 | Terrace/Linked House | 46.0 |
| Pre university/Undergraduate | 77.0 | Flat/Apartment/Condominium | 2.0 |
| Postgraduate | 17.0 | Shop House | 2.0 |
| Others | 2.0 | Others | 0.0 |
| Living Companion | | Housing Ownership | |
| Family | 95.0 | Rent | 7.0 |
| Friend(s) and Acquaintance(s) | 4.0 | Own | 83.0 |
| Others | 1.0 | Other | 10.0 |

**Table 2.** Waste knowledge level among respondents.

| Knowledge Level | General Knowledge | | | Specific Knowledge | | |
|---|---|---|---|---|---|---|
| | High | Moderate | Low | High | Moderate | Low |
| Range of Score | 10 to 8 | 7 to 5 | 4 to 0 | 10 to 8 | 7 to 5 | 4 to 0 |
| Frequency (%) (*n* = 100) | 37.0 | 59.0 | 4.0 | 6.0 | 65.0 | 29.0 |

**Table 3.** General knowledge level among respondents.

| Statement (*n* = 100) | True (%) | False (%) | Correct Answer |
|---|---|---|---|
| 1. Recycling and waste sorting at source cannot help to curb insufficient landfill capacity. | 25.0 | 75.0 | False |
| 2. Recycling and waste sorting can help to reduce the spread of disease like bacterial or fungal infections. | 82.0 | 18.0 | True |
| 3. Waste sorting can prevent emission of harmful chemicals and greenhouse gasses (methane and carbon dioxide) that contribute to global warming. | 94.0 | 6.0 | True |
| 4. Waste without sorting can be used to create compost for soil fertility regeneration. | 35.0 | 65.0 | True |
| 5. Energy used to make a new product from raw materials is far less than energy required for recycling. | 45.0 | 55.0 | False |
| 6. Waste sorting can prevent contamination of recyclables. | 93.0 | 7.0 | True |
| 7. More waste fractions need to be dealt with when practising waste sorting than recycling. | 84.0 | 16.0 | True |
| 8. Recyclables collected can only be recycled once. | 26.0 | 74.0 | False |
| 9. All plastics that contain numbered symbols (also known as plastic resin identification codes) can be recycled. | 71.0 | 29.0 | False |
| 10. Waste is sorted and collected at household, but not necessarily recycled. | 64.0 | 36.0 | True |

**Table 4.** Specific knowledge level among respondents.

| Selected Waste Item (*n* = 100) | 1. Biodegradable/Organic Waste (%) | 2. Non-Biodegradable/Non-Organic Waste (%) | 3. Recyclable Waste (%) | 4. Hazardous Waste (%) | Correct Waste Category |
|---|---|---|---|---|---|
| 1. Used Tissue Paper | 52.0 | 23.0 | 23.0 | 2.0 | 2 |
| 2. Food Stained Paper or Plastic Container | 14.0 | 55.0 | 27.0 | 4.0 | 2 |
| 3. Light Bulb | 7.0 | 40.0 | 14.0 | 39.0 | 4 |
| 4. Vegetable and Fruit Peel | 93.0 | 2.0 | 3.0 | 2.0 | 1 |
| 5. Bone and Meat Scrap | 89.0 | 8.0 | 2.0 | 1.0 | 1 |
| 6. Dry Leaves | 85.0 | 10.0 | 5.0 | 0.0 | 1 |
| 7. Expired Medicine or Supplement | 19.0 | 35.0 | 3.0 | 43.0 | 4 |
| 8. Photograph and Paint | 7.0 | 51.0 | 13.0 | 29.0 | 4 |
| 9. Electronic, Batteries and Cables | 6.0 | 36.0 | 14.0 | 44.0 | 3 |
| 10. Hair or Fur | 53.0 | 36.0 | 9.0 | 2.0 | 1 |

Table 4 shows the responses to the specific knowledge of R&SS based on waste categories for selected items. Most of the respondents selected the categories of waste correctly, except for waste items #1: used tissue paper, #8: photographs and paint, and #9: electronics, batteries, and cables.

### 3.3. Attitude Assessment

Overall, the attitude score among the respondents was relatively positive compared to the knowledge score, where 75.0% of the respondents scored between 52 and 70 (Table 5).

**Table 5.** Attitude level among respondents.

| Attitude Level | Positive | Neutral | Negative |
|---|---|---|---|
| Range of Score | 70 to 52 | 51 to 33 | 32 to 14 |
| Frequency (%) (*n* = 100) | 75.0 | 19.0 | 6.0 |

Table 6 reveals the results about respondents' perceived participation in different stages of waste management, starting from waste generation, waste storage, waste collection, waste processing, and waste disposal. Overall, the respondents demonstrated a high level of intention to participate in waste management processes, except for the waste collection stage, specifically statement #6: I am willing to pay extra service charges for different waste collection according to its category, and statement #8: I am actively involved in the collection and transport of sorted waste materials in my neighbourhood.

**Table 6.** Involvement in waste management stages.

| Level of Importance on Statement (*n* = 100) | 1. Not Important at all | 2. Slightly Important | 3. Moderately Important | 4. Very Important | 5. Extremely Important | Mean | Standard Deviation |
|---|---|---|---|---|---|---|---|
| Importance of Waste Separation as Immediate Solution to Waste Crisis | 1.0 | 2.0 | 14.0 | 44.0 | 39.0 | 4.18 | 0.817 |

| Level of Agreement on Statement (*n* = 100) | 1. Strongly Disagree | 2. Disagree | 3. Neutral | 4. Agree | 5. Strongly Agree | Mean | Standard Deviation |
|---|---|---|---|---|---|---|---|
| **Waste Generation** | | | | | | | |
| 1.    I am responsible for the waste I generated | 5.0 | 0.0 | 10.0 | 49.0 | 36.0 | 4.11 | 0.948 |
| 2.    The purchase decisions that I make can increase or decrease the amount of waste I need to get rid of. | 6.0 | 0.0 | 15.0 | 47.0 | 32.0 | 3.99 | 1.005 |
| 3.    I feel satisfied when waste is sorted and can be a resource. | 6.0 | 2.0 | 12.0 | 45.0 | 35.0 | 4.01 | 1.044 |
| Average Weighted Mean | | | | | | 4.03 | Agree |
| **Waste storage** | | | | | | | |
| 1.    I play an important role in sorting waste within my household | 5.0 | 6.0 | 18.0 | 38.0 | 33.0 | 3.88 | 1.089 |
| 2.    I am willing to separate waste into respective category before collection | 4.0 | 1.0 | 23.0 | 41.0 | 31.0 | 3.94 | 0.968 |
| Average Weighted Mean | | | | | | 3.91 | Agree |
| **Waste Collection** | | | | | | | |
| 1.    I am willing to pay extra service charges for different waste collection according to its category | 10.0 | 14.0 | 26.0 | 38.0 | 12.0 | 3.28 | 1.150 |
| 2.    I am satisfied with having different types of waste being transported to their respective site. | 7.0 | 1.0 | 20.0 | 49.0 | 23.0 | 3.80 | 1.030 |
| 3.    I am actively involved in collection and transport of sorted waste materials in my neighbourhood | 5.0 | 25.0 | 35.0 | 25.0 | 10.0 | 3.10 | 1.044 |
| Average Weighted Mean | | | | | | 3.39 | Neutral |
| **Waste processing** | | | | | | | |
| 1.    I am willing to participate in training or programmes to gain knowledge on correct waste sorting methods. | 3.0 | 9.0 | 36.0 | 37.0 | 15.0 | 3.52 | 0.954 |
| 2.    Waste sorting and disposal should be taught in school as part of environmental education. | 5.0 | 2.0 | 7.0 | 40.0 | 46.0 | 4.20 | 1.010 |
| 3.    The municipal council is not doing enough to fix the garbage problem. | 8.0 | 2.0 | 20.0 | 36.0 | 34.0 | 2.14 | 1.149 |
| Average Weighted Mean | | | | | | 3.86 | Agree |
| **Waste Disposal** | | | | | | | |
| 1.    I do not care that burning garbage can harm my health and the health of others. | 60.0 | 21.0 | 7.0 | 9.0 | 3.0 | 1.74 | 1.110 |
| 2.    People throw garbage on the streets and in the drains because they have no other choice to get rid of the garbage. | 53.0 | 24.0 | 9.0 | 10.0 | 4.0 | 1.88 | 1.169 |
| Average Weighted Mean | | | | | | 1.81 | Disagree |

### 3.4. Behaviour Assessment

Unlike the scores in the categories of knowledge and attitude, the scores in the behaviour category among the respondents tended towards negative, with a score of 68.0% (Tables 7 and 8).

**Table 7.** Behaviour level among respondents.

| Behaviour Level | Good | Satisfactory | Poor |
|---|---|---|---|
| Range of Score | 10 to 8 | 7 to 5 | 4 to 0 |
| Frequency (%) ($n = 100$) | 4.0 | 28.0 | 68.0 |

**Table 8.** Methods of waste disposal among respondents.

| Selected Waste Item ($n = 100$) | 1. Dumping (Garbage Bins, etc.) | 2. Illegal Dumping (Forest, River) | 3. Open Burning | 4. Feeding to Livestock or Pets | 5. Using Compost Pits or Burying | 6. Recycling after Cleaning | 7. Sending to Collection Centre | Acceptable Waste Disposal Method (%) |
|---|---|---|---|---|---|---|---|---|
| 1. Used tissue paper | 87.0 | 2.0 | 0.0 | 0.0 | 1.0 | 3.0 | 7.0 | 87.0 |
| 2. Food-stained paper or plastic container | 58.0 | 1.0 | 1.0 | 3.0 | 5.0 | 26.0 | 6.0 | 26.0 |
| 3. Light bulb | 68.0 | 4.0 | 0.0 | 0.0 | 3.0 | 3.0 | 22.0 | 25.0 |
| 4. Vegetable and fruit peel | 46.0 | 2.0 | 1.0 | 17.0 | 33.0 | 0.0 | 1.0 | 51.0 |
| 5. Bone and meat scrap | 47.0 | 2.0 | 1.0 | 32.0 | 17.0 | 0.0 | 1.0 | 50.0 |
| 6. Dry leaves | 45.0 | 3.0 | 10.0 | 1.0 | 39.0 | 1.0 | 1.0 | 41.0 |
| 7. Expired medicine or supplement | 80.0 | 1.0 | 0.0 | 0.0 | 4.0 | 3.0 | 12.0 | 12.0 |
| 8. Photograph and paint | 71.0 | 0.0 | 1.0 | 0.0 | 1.0 | 8.0 | 19.0 | 19.0 |
| 9. Electronic, batteries and cables | 55.0 | 0.0 | 1.0 | 0.0 | 3.0 | 6.0 | 35.0 | 41.0 |
| 10. Hair or fur | 84.0 | 2.0 | 2.0 | 0.0 | 8.0 | 2.0 | 2.0 | 12.0 |

Note: Each item can have one or more than one acceptable waste disposal method. Only those highlighted in grey was added and given 1 score each.

When the relationship between demographic characteristics and KAB was tested, the Chi-Square test revealed a significant relationship between household size and general knowledge and between housing type and attitude ($p$ value < 0.05) (Table 9).

Further analysis using Pearson's Correlation Coefficient at a significant level of a = 0.05 was conducted to test the relationship between KAB. Only general knowledge and specific knowledge demonstrated a moderate relationship, with some nearly negligible relationships present between specific knowledge and attitude, as well as with behaviour (Table 10).

**Table 9.** Relationship between demographic characteristics and KAB.

| Demographic Characteristics (*n* = 100) | General Knowledge (*n*; %) | | | Total (%) | *p*-Value |
| --- | --- | --- | --- | --- | --- |
| | Low | Moderate | High | | |
| Household Size | | | | | |
| Alone | 0; 0.0 | 2; 66.7 | 1; 33.3 | 3.0 | |
| 2 to 4 | 2; 5.2 | 21; 53.8 | 16; 41.0 | 39.0 | |
| 5 to 7 | 1; 1.8 | 35; 62.5 | 20; 35.7 | 56.0 | 0.046 |
| More than 7 | 1; 50.0 | 1; 50.0 | 0; 0.0 | 2.0 | |
| Total | 4.0 | 59.0 | 37.0 | 100.0 | |
| **Demographic Characteristics (*n* = 100)** | **Attitude (*n*; %)** | | | **Total (%)** | **_p_-Value** |
| | Negative | Neutral | Positive | | |
| Housing Type | | | | | |
| Bungalow/Detached House | 0; 0.0 | 3; 13.6 | 19; 86.4 | 22.0 | |
| Semi-Detached House | 4; 14.3 | 3; 10.7 | 21; 75.0 | 28.0 | |
| Terraced/Linked House | 1; 2.1 | 13; 28.3 | 32; 69.6 | 46.0 | 0.027 |
| Flat/Condominium | 1; 50.0 | 0; 0.0 | 1; 50.0 | 2.0 | |
| Shop House | 0; 0.0 | 0; 0.0 | 2; 100.0 | **2.0** | |
| Total | 6.0 | 19.0 | 75.0 | 100.0 | |

**Table 10.** Correlation between KAB.

| Variables | *p*-Value | R-Value |
| --- | --- | --- |
| General Knowledge and Specific Knowledge | $4.90 \times 10^{-19}$ | 0.342 |
| General Knowledge and Attitude | $7.39 \times 10^{-71}$ | 0.049 |
| General Knowledge and Behaviour | $2.37 \times 10^{-24}$ | 0.059 |
| Specific Knowledge and Attitude | $7.12 \times 10^{-73}$ | 0.107 |
| Specific Knowledge and Behaviour | $6.79 \times 10^{-9}$ | 0.104 |
| Attitude and Behaviour | $3.61 \times 10^{-73}$ | 0.021 |
| Knowledge and Attitude | $7.30 \times 10^{-66}$ | 0.095 |
| Knowledge and Behaviour | $8.43 \times 10^{-49}$ | 0.099 |

*3.5. Participation in Recycling and Waste Separation Practices*

A total of 62% respondents indicated that they were involved in some extent of recycling practice, while 28.0% practised waste separation. The Chi-Square test also indicated the significance between recycling and source separation practices, where about two-fifth of recyclers adopted source separation (Table 11). In addition, a significant association between demographic profile and source separation practice was revealed through the Chi-Square test. These included age group, marriage status, educational level, and duration of stay (Table 12).

**Table 11.** Association between recycling and source separation participation.

| Recycling Participation (*n* = 100) | Source Separation Participation (*n*; %) | | Total (%) | *p*-Value |
| --- | --- | --- | --- | --- |
| | Yes | No | | |
| Yes | 25 (40.3) | 37 (59.7) | 62.0 | |
| No | 3 (7.9) | 35 (92.1) | 38.0 | 0.046 |
| Total | 28.0 | 72.0 | 100.0 | |

**Table 12.** Association between demographic profile and source separation participation.

| Demographic Characteristics (*n* = 100) | Source Separation Participation (*n*; %) | | Total (%) | *p*-Value |
|---|---|---|---|---|
| | **Yes** | **No** | | |
| Age Group | | | | |
| 18 to 24 years old | 17 (29.3) | 54 (76.1) | 71.0 | |
| 25 to 44 years old | 8 (30.8) | 18 (69.2) | 26.0 | 0.046 |
| 45 to 64 years old | 3 (100.0) | 0 (0.0) | 3.0 | |
| Total | 28.0 | 72.0 | 100.0 | |
| Marital Status | | | | |
| Single | 21 (24.1) | 66 (75.9) | 87.0 | 0.026 |
| Married | 7 (53.8) | 6 (46.2) | 13.0 | |
| Total | 28.0 | 72.0 | 100.0 | |
| Educational Level | | | | |
| Upper Secondary | 0 (0.0) | 4 (100.0) | 4.0 | |
| Pre-university | 2 (50.0) | 2 (50.0) | 4.0 | |
| Diploma | 0 (0.0) | 3 (100.0) | 3.0 | |
| Bachelor | 15 (21.4) | 55 (78.6) | 70.0 | 0.027 |
| Postgraduate | 4 (44.4) | 5 (55.6) | 9.0 | |
| Master | 5 (62.5) | 3 (37.5) | 8.0 | |
| Other | 2 (100.0) | 0 (0.0) | 2.0 | |
| Total | 28.0 | 72.0 | 100.0 | |
| Duration of Stay | | | | |
| Less than a year | 1 (25.0) | 3 (75.0) | 4.0 | |
| 1 to 5 years | 7 (77.8) | 2 (22.2) | 9.0 | |
| 6 to 10 years | 4 (36.4) | 7 (63.6) | 11.0 | |
| 11 to 15 years | 2 (28.6) | 5 (71.4) | 7.0 | 0.018 |
| 16 to 20 years | 4 (23.5) | 13 (76.5) | 17.0 | |
| More than 20 years | 10 (19.2) | 42 (80.8) | 52.0 | |
| Total | 28.0 | 72.0 | 100.0 | |

Tables 13–16 show the statements asked in Section E of the questionnaire, where each respondent was able to choose multiple answers (reason statements) resulting in varying numbers of frequency recorded across the tables. Protecting the environment was rated as the most motivating reason to participate in recycling, consisting of about one quarter of the total frequency. That the ranking was immediately followed by the feeling of satisfaction in anticipating that waste would become a new resource and their continuous effort would influence other household members and neighbours as the reasons for recycling, whereas the skill of handling and sorting the recyclables was rated as the least important reason to participate in recycling (Table 13).

Three reasons voted the most by the majority of the respondents for practising waste separation are as follows (Table 14):

- It provides more environmental benefit than recycling (to prevent hazardous waste from polluting the landfills).
- It is important to separate biodegradable waste with recyclables to prevent contamination.
- It is more rewarding than recycling because biodegradable waste can be used as other resources (to feed animals, use as soil fertiliser for crops, etc.).

**Table 13.** Reasons to participate in recycling.

| Statement | Frequency (%) (*n* = 114) |
|---|---|
| 1. It is very convenient to recycle where I live. | 9.6 |
| 2. I have spare time to sort out recyclables from the general waste. | 7.0 |
| 3. Recycling is very cost rewarding (monetary or physical item). | 10.5 |
| 4. I am good at handling and sorting recyclables. | 4.4 |
| 5. I believe my continuous recycling effort will influence my household members and neighbours. | 14.0 |
| 6. I feel satisfied when waste becomes a new resource. | 15.8 |
| 7. I practise recycling because it can protect the environment. | 26.3 |
| 8. It is a habit I developed from my upbringing. | 12.3 |
| 9. Others. | 0.0 |

**Table 14.** Reasons to participate in source separation.

| Statement | Frequency (%) (*n* = 83) |
|---|---|
| 1. It provides more environmental benefit than recycling (to prevent hazardous waste from polluting the landfills). | 22.9 |
| 2. It is important to separate biodegradable waste with recyclables to prevent contamination. | 24.1 |
| 3. It is more rewarding than recycling because biodegradable waste can be used as other resources (to feed animals, use as soil fertiliser for crops, etc.). | 16.9 |
| 4. Convenient (collected and transported to collection centre). | 13.3 |
| 5. Having more spare time. | 6.0 |
| 6. Good at sorting different waste. | 6.0 |
| 7. An extension from recycling habits. | 10.8 |
| 8. Others. | 0.0 |

**Table 15.** Reasons to not participate in recycling.

| Statement | Frequency (%) (*n* = 59) |
|---|---|
| 1. It is not accessible or convenient to where I live. | 16.9 |
| 2. It takes up too much time (to clean out/prepare recyclables, to travel to the nearest recycling centre, to look for information regarding each waste fraction). | 20.3 |
| 3. I always forget. | 13.6 |
| 4. Cost over benefit (where the reward from recycling does not feel worthy which could not cover the time consumed or transportation cost, and storage cost). | 3.4 |
| 5. I am not sure which waste belongs to which category. | 11.9 |
| 6. I do not feel my recycling efforts will make a difference. | 6.8 |
| 7. I feel uncomfortable having many recyclables bins or bags for different waste categories in my household. | 8.5 |
| 8. I do not care about recycling as it is not my priority for environmental concern. | 1.7 |
| 9. I do not understand the environmental benefit that recycling can provide. | 3.4 |
| 10. None of my household members recycle. | 13.6 |

**Table 16.** Reasons to not participate in source separation.

| Statement | Frequency (%) (*n* = 174) |
|---|---|
| 1. It is more tedious than recycling. | 9.8 |
| 2. There is no collection service for each type of waste even if I sort it out. | 22.4 |
| 3. There is no monetary reward unlike recycling. | 5.2 |
| 4. It does not bring more environmental benefit than recycling. | 1.7 |
| 5. Not convenient. | 18.4 |
| 6. More time needed. | 16.7 |
| 7. More knowledge needed. | 19.0 |
| 8. My household members always ruin my sorting efforts. | 6.9 |
| 9. Others. | 0.0 |

When asked about the reasons for not practising recycling, time consuming was ranked as the most important consideration. Absence of an influencer at home, issues of convenience, and accessibility to the recycling/collection centre, as well as the lack of knowledge of recyclable wastes were also among the most selected reasons that set the barriers to recycling practice among the respondents (Table 15). A similar ranking of reasons (except for no home influencer) was revealed by respondents who did not participate in waste separation (Table 16).

When asked about action taken when facing uncertainty regarding the waste category, most of the respondents would place the item in the common trash and only a handful would refer to the online resources for reference (Table 17).

**Table 17.** Action taken upon uncertainty of waste category.

| Action | Frequency (%) (*n* = 100) |
|---|---|
| 1. Place the item in the trash. | 54.0 |
| 2. Place the item in the recycling bin. | 7.0 |
| 3. Refer to the available guide from the local authority. | 8.0 |
| 4. Contact waste collection service provider for advice. | 0.0 |
| 5. Refer to someone who you think has more knowledge on waste sorting. | 4.0 |
| 6. Refer to online resources. | 27.0 |
| 7. Others. | 0.0 |

*3.6. Perspectives of R&SS Service Users*

All 7 interviewees from the service users' group (including non-recyclers or non-waste sorters) held a positive attitude towards the products and compost made from recycled material. They had a satisfactory level of awareness and were able to explain the advantages and disadvantages of the waste treatment from R&SS. Nevertheless, the disadvantages for waste separation per se relatively outweighed and discouraged them from the practice, as opposed to that of recycling (Table 18). All 7 interviewees confirmed that they had no knowledge of the existing R&SS related efforts or initiatives implemented by Manjung Municipal Council and any local non-governmental organizations.

**Table 18.** Interview responses from the perspective of service users.

| No. | Questions/Matters Discussed |
|---|---|
| 1. | The interviewees were asked if they knew of or had seen any product made from recycled materials to confirm the outcomes of recycling itself and whether their recycling efforts were making an impact [39]. |
| Responses from Interviewees | As one of the respondents said:<br>"I have no idea how the recyclables I have collected are treated at the end of the process. It keeps me doubtful if truly these recyclables are transformed into useful raw materials and whether my recycling effort makes a significant impact. At least, products made from recyclables are not commonly seen in the market yet and can only be found in shops that are designed to sell such products. Usually, sustainable products that I encounter are marked "organic" rather than "made from recycled material". |
| 2. | The interviewees were asked if they knew of or had seen (had experience) compost, a product generated through recycling food waste (sorted from general waste) to confirm the level of willingness to participate in the process from waste sorting to compost making, and in addition to this, they were asked about their challenges or gains to participate in this process [39]. |
| Responses from Interviewees | In terms of composting practice (one of the outcomes from source separation), this was what one respondent said:<br>"Composting allows a complete cycle of waste management. It can self-sustain where the biodegradable waste generated can be turned into fertiliser for plants that produce food. However, due to the lack of space (land for farming) and lack of influence from the households' waste handler, composting is not what I can usually practice at home. It is less useful to me compared to other households that have the supporting mechanism and medium (e.g., planting ground where compost is valuable) to do so".<br>Another respondent added:<br>"Composting is not a practice that households normally do. This is because it requires a tedious procedure, which is to ensure the right composition of carbon and nitrogen. Without the proper procedure, it will produce odour and attract pests and insects. Even the easiest composting method, the Bokashi, is also an unpleasant process during the accumulation of 9-litre kitchen waste in a week. This is because the process is not as straightforward as it seems, and it is generally harder to control. It might contain materials that are not advisable to compost, like oil or liquid, meat, bones, and dairy. A more complete composting system that includes a wide range of categories is more expensive at a point that common households cannot afford or are not willing to pay. For example, the composting machine from MAEKO, even though it is designed for households, it is still not considered as cost-efficient for common households' ability to pay". |
| 3 | The interviewees were further asked if they could consider other waste minimisation options (moving up the waste hierarchy [12]). |
| Responses from Interviewees | In response to the difficulties in R&SS, a respondent also commented on waste reduction and where his priority lies:<br>"Going zero waste is very difficult when my priority does not lie on extreme waste minimisation as it is not cost-effective based on my affordability for essential purchases—groceries or daily necessities. This also requires strong determination that I do not think is worthy for me to trade off if there is no collective effort from a large population". |

## 4. Discussion

The population in Manjung is surging, becoming the third most populated district in Perak. The increasing volume of waste generated as a result of throw-away consumerism will eventually exceed the capacity of the waste treatment at the only landfill site in Manjung, whose lifespan is expected to end within 3 to 5 years (data from Manjung Municipal Council). The heavy reliance on landfill for waste disposal has created a comfortable ground for households to throw every type of waste into the garbage bin for weekly municipal waste collection. R&SS therefore remains an unpopular option, and this situation

is not acceptable for a city striving for sustainability within the next decade. This study investigates the gap between households' KAB that has delayed the transition towards sustainable integrated waste management. In addition to the gap, households' behaviour is also influenced by potential parameters that not only radiate from their mental characteristics (internal factors) but are also shaped by the physical environment and on how they deal with waste, from consumption to disposal (external factors). The causation of households' behavioural pattern is to be discussed in this section and should be addressed with relevant improvements in the waste management service to enhance the sustainability of Manjung district.

### 4.1. Associations between the Households' Level of KAB towards R&SS Practices

The relationship between knowledge, attitude, and behaviour is direct in theory, where knowledge supports the foundation of information processing, attitude bears psychological and emotional development, and behaviour holds the response and reaction through the entire process of thinking, feeling, and acting [23,46]. However, KAB relate to each other differently and inconsistently in reality, resulting in gaps between familiarity, values, and actions [47].

This gap in KAB also exists in the waste management system in Manjung in regard to whether the local authority or state government should implement Act 672 for mandatory source separation, based on the current condition and quality of the system. In reality, the recyclables, which have been produced and consumed, remain mismatched with the actual processed recyclables. Households also find it difficult to commit to recycling alone (less than 1.0% recycling rate compared to total waste generated per year in Manjung), not to mention source separation, due to additional requirements.

The findings show that households' knowledge (general and specific) is at a moderate level, attitude is at a positive level, and behaviour is at a poor level. These results denote that these three components influence and can be influenced by each other reciprocally, as suggested by various studies in the literature, if not direct. On the other hand, even though KAB has no eligible linear associations statistically, their associations are significant when discussed with multiple sets of conditions, variables, or causal factors. In fact, the negligible linear associations between these three components have proven that their associations are rather complex [46]. Similar results have been recorded in several studies; for instance, ref. [48] agree to the complexity in the study of KAB, especially for the fact that knowledge and attitude for participation in "green activities" do not lead directly to behaviour, regardless of the level. They have discussed that their respondents are familiar with the idea of recycling, yet the practical aspect is absent due to their indifferent attitudes towards the practice.

Finally, many scholars have also studied the associations between demographic profile and the KAB components. In this study, the findings only show household size and housing type as having statistically significant associations with one of the components. In fact, ref. [22] describe the associations of a high satisfactory level of KAB with a medium-sized household where the adults (parents) possess high educational level and secure jobs. Landed property has an influence on households' attitude on the local waste management system rather than knowledge, as suggested by [30]. Other demographic characteristics such as age, gender, civil status, and income level are also major determinants in establishing associations with KAB components [30,48]. Similarly, age, civil status, educational level, and living duration in this study show statistically significant associations with households' participation in source separation practices, which also influences the reciprocal causation of KAB components.

### 4.2. Enabling Factors of Households' Behaviour on R&SS Practices

Sections 4.2 and 4.3 discuss the understanding of the households' behavioural pattern (based on the sample at the point of observation, survey, and interview) with the identified barriers and enabling factors that are categorised into internal and external

aspects. These parameters can bridge the gap of the KAB and serve as a basis to formulate recommendations. According to the findings, the internal factors are boiled down to (a) intentions, (b) knowledge, (c) willingness, (d) habit or routine, together with (e) persuasion. Conversely, the external factors identified in this study for the waste management service include (i) accessibility to services (both storage and collection), (ii) tangible incentives (reward) and disincentives (enforcement of law and penalty), and (iii) restrictions on consumption (discouragement of throw-away consumerism).

In terms of enabling factors, the findings (survey and elicited data) reveal that environmental protection is the fundamental motivation for the participation of R&SS practices among the respondents. The strength of intentions propels the recyclers (about 40%) to also participate in source separation, even though they have to deal with inconvenience within the provided waste management system. Households would have to take their own initiative to transport sorted waste to the collection centre. In fact, the municipal council only provides a limited number of recycling containers (a total of 11) and they are dedicated to collect specific and narrow waste category. This has lowered the willingness level, as it was ranked rather low as enabling factors for households to participate in R&SS. To increase this willingness, it may have to couple with other enabling factors such as tangible incentives (reward) and high accessibility to collection services with a widely acceptable waste category, as suggested by [33,39].

Knowledge, in this context, refers to the implications of R&SS practices and knowing the results from their participation in R&SS [35,39]. Interviewees gave positive responses towards the publicity of the outcomes of waste treatment through R&SS, which they claimed would greatly encourage them as they could know how their recycling effort counts. Having this knowledge in mind, this could have strengthened the participation more as "the feeling of satisfaction for anticipating that waste would become a new resource" and "continuous effort" are among the enabling factors that are rated high in the ranking.

Finally, Malaysia has a long recycling implementation history [7] and, consequently, this habit influences the decisions of households [24]. At least 10% of the respondents engaged in R&SS, behaviour that has been partially enabled by recycling habits. Based on the elicited data, songs have been created to teach children how to differentiate the colours of the recycling bin for each type of waste; such methods prompt children to engage in sustainable processes, for instance, paper must go into the blue recycling bin (when you see one).

Generally, this part of the discussion could assist service providers, especially the local authorities, to prioritise their agenda and budget to facilitate the transfer of knowledge among households [35,39].

### 4.3. Barriers of Households' Behaviour on R&SS Practices

Willingness to invest in terms of time, money, and effort was lacking among households (reasons ranked amongst the highest at first in recycling; fourth in source separation) as they have been practising a more "convenient" way of consumption and waste management for the past few decades; for example, taking single-use plastics for granted and dumping all wastes into garbage bins [37]. A change in lifestyle is difficult, as claimed by the majority of the interviewees. Dumping, which is a traditional (conventional) practice, is easier to continue as a habit compared to recycling and composting, which require gaining new knowledge and taking more steps to apply [49]; knowledge was rated as the second most selected barrier to participation in source separation.

In fact, there is a disconnection between households' pro-environmental behaviours (acted upon a just cause) and the final outcomes of the R&SS practices; when the interviewees were asked if they knew how recycling could protect the environment, most of them were only able to guess that the recyclables collected are being transformed into secondary material for production. They were intrigued to know the outcomes of the items they recycled; however, this information is not made available.

In fact, the waste management system (current R&SS service) in Manjung operates without a responsive feedback loop to keep users informed and educated. There are limited channels for respondents to improve their waste sorting knowledge, rebut recycling myths, track their contribution in R&SS practices, as well as follow up the aftermath of the segregated waste after it has been collected. The lack of such information discourages continuous recycling efforts among households, especially in communities where the idea that "waste that has been put in a recycle bin does not mean that it has been actually recycled" is a common belief [50,51]. This feedback is essential to prevent "wish-cycling": irresponsibly placing items into a recycle bin and hoping that it would be recycled [52,53]. The discouragement also presents a negative influence towards the impact of one person recycling, as the lack of communal effort has widely become an excuse (ranked as the third most selected barrier in recycling) for most households (not convinced without actual and strong evidence) to refuse in taking up these practices, especially source separation.

Generally, source separation is more complex than recycling, which has caused the percentage of participation to drop to 28% from that of recycling at 62%. It therefore depends on the degree of the discussed internal factors, which can determine how poor a household's behaviour is, in what ways their behaviour is poor, and to what extent this behaviour can be improved.

In terms of external factors, the findings (observed and elicited data) show that all waste management stages in Manjung pose challenges to households' participation in R&SS, especially source separation. As observed, households' waste generation is aggravated by the culture of throw-away consumerism and the convenience of plastic usage in the commercial industry. Waste is generated rapidly not only due to the increasing population (overconsumption), but also the short life cycle of the purchased product (made from non-decomposable material) [3,6]. Many are contaminated by materials such as food residue without proper sorting when discarded [36,52]. Additionally, the district has adopted a waste storage system that does not encourage the action of segregation due to its one-type design, especially waste containers provided at public areas (e.g., wet market). For waste collection, most housing developments in Manjung are landed property, where curb side recycling is claimed to have an overall positive effect for source separation [54], yet this idea is not celebrated. Integrating source separation with recycling seems impossible, as almost all local recycling vendors do not accept biodegradable waste and waste with mixed material (e.g., milk carton). Moreover, recyclables are usually self-transported, and the accepted recyclable categories are limited and generally remain unclear to most of the recyclers. A total of 40.3% of the respondents stated that they will still engage in both R&SS practices even though they have not been well supported by the current waste management system and environment. This could probably be due, in part, to their pro-environmental behaviour. However, at the same time, accessibility to R&SS services is also one of the greatest challenges for them and might result in negative experience when engaging in R&SS in the long term. This is because accessibility to R&SS services has been rated as the second most important barrier to participation in source separation by most of the non-recyclers and non-waste sorters. This barrier has stronger implications for service users in R&SS, discouraging participation.

The interviewees generally feel negatively towards the use of plastic bags, as they understand that the ocean and marine life are impacted by this consumption (based on elicited data). However, when the use of plastic bags grants them great convenience, they still decide to compromise. This is described as selective empathy, in which people in general selectively care about one matter instead of the actual problem [55]. This widens the mismatch between the psychology of households and their behaviour, where mental characteristics are not strong enough to effectively drive a favourable outcome due to the lack of a support system from waste management mechanisms at all stages. This support system can create a convenient environment for households to easily partake in R&SS [33,39]. This convenience is totally different from the convenience associated with throw-away consumerism. To date, a series of programmes have been implemented in

Manjung to encourage R&SS participation at the household level. These include (a) education programmes for pupils from kindergarten, primary, and secondary school. A total of 70 kindergartens have been engaged throughout the year to instil recycling habits in children's upbringing. Recycling and waste separation habits are generally more celebrated among school children; (b) engagement programmes with a group of households to adopt composting (Takakura method) for biodegradable waste treatment. This compost is then used for plantation on the land provided by the local authority (Department of City Planning), named as Taman Communiti, in a planned neighbourhood: Phase 1D Manjung. The community has been taught to produce their own compost after several tutorials; (c) monthly recyclables trade-off programmes for daily necessities at the main lobby of Manjung Municipal Council Office Building every first Saturday of the month; (d) provision of recycling containers (a total of 11) at public areas to collect textiles from households, known as SULAM programmes; (e) a pilot project with business owners at public wet markets with the aim of halting single-use plastics for packaging; and (f) focus group discussions (shifted from pupils to families) in various neighbourhoods.

Through these existing initiatives, the local authority has noticed the difficulty in reaching out or to convince the diverse population within the district to participate in their R&SS implementation. Information is usually disseminated through online platforms—official portals and social media (Facebook) page—as well as billboard advertisements at two of the busiest crossroads. Yet, the desirable results have been difficult to achieve. The lack of effective engagement and communication strategies and long-term trust building with the community are among the major issues faced by the local authority [56,57]. The interviewee from the local authority admitted that they were hoping that NGO(s) would come to them with a R&SS implementation proposal so that they could provide these NGO(s) with resources.

In addition, the State's commitment to sustainable waste management has been nonchalant and, as a result, the local authority's initiatives have only been sufficient enough to fulfil the requirements of standard urban management rather than a mission. Additionally, being a suburban municipality, the economy, technology, and even society are still unprepared for the development of sustainable waste management [47]; the communities in Manjung have yet to emerge as a major driving force behind the waste minimisation movement.

*4.4. Recommendations to Enhance R&SS Participation*

Considering all parameters discussed in 4.2. and 4.3., the authors made the following recommendations:

In order to increase the willingness of households to participate in sustainable practices, an education programme with an effective feedback loop system via internet communication technology is essential [17]. There are a few key elements in this system. First, a reactive feedback platform (online message, calls, and emails) answers to any uncertainty during waste sorting—a live version of gomi (Japanese word of "garbage") guide—becoming an ultimate go-to reference for waste sorting. This can improve households' perception of how easy R&SS can be carried out. Second, concise and practical information or knowledge is fed through push notifications to spark smaller actions. This further allows households (especially waste handlers) to react positively towards R&SS, as small actions do not incur heavy costs. Lastly, networking platforms allow households to share success stories and motivation. This can cultivate positive peer pressure and redefine subjective norms of the community to manifest the fact that their effort can make a difference.

Meanwhile, religious and dialect associations are groups that could bring households together, mobilise them and empower them to accomplish a mission. Households that are relatively able to placed their trust in these associations and dialogue can be easily initiated when two groups share similar social-cultural backgrounds [58]. Furthermore, as demonstrated by the community in Pulau Pangkor through authors' observation and interviews with the local NGO, a deep trust is the key driving factor for the third party to be able to encourage better participation in R&SS practices among households. This trust

is formed when people share a common vision through long-term communication and engagement. By doing so, it allows both parties to easily unite under a common interest or goal when they have achieved the same ideology. A higher success rate is guaranteed with stronger social capital and creative social innovation [58]. This social innovation is important for the community to urge local authority to facilitate an enabling environment for R&SS implementation.

Environmental activities such as recycling and composting are best started off from these associations while utilising their facilities, as they have set up optimum conditions for community-driven projects. For instance, in Sentul, Kuala Lumpur, a Sikh community realised that a huge amount of biodegradable waste was being generated through their religious activities. This encouraged them to initiate a composting movement in reusing food and garden waste around their neighbourhood, while at the same time utilising the space around its religious facility [59]. It has also attracted families to join the action to contribute to waste reduction.

Regarding this example, the local authority can be the intermediary to connect the community with various sectors that require R&SS services or products. For instance, the compost produced by the community can be sold to agriculture-related sectors (e.g., farm or plantation within or outside the district) via the local authority. The local authority can also take up a role as an enabler to provide various resources, including space and location, information such as waste composition and waste handling within a community, as well as regulations such as legislative by-laws and guidelines for proper practice and implementation. Merits or incentives should be introduced to those who make a significant impact on minimising the waste, thus reinforcing their actions and advocating change.

The outcomes of R&SS should be widespread, providing both environmental as well as economic benefits. Households should be constantly reminded of this knowledge to shape their attitude slowly but surely. Waste management is a service delivery; the expenditure should be wisely spent to achieve a more productive result within a limited budget. Therefore, it is important to let households know that R&SS implementation can significantly reduce the amount of waste that has been sent to landfills, increasing the lifespan of landfills, and thus saving municipal budgets.

Furthermore, the authors in [60] have revealed that it is important for local authorities to stay financially feasible and sustainable through cost recovery, such as from transportation cost saving through lesser intervals as less generated waste is being sent to landfill. The authors in [61] support this idea by suggesting the R&SS industry collaborate with local authorities to improve the efficiency of recycled waste processing. Cost recovery can also be carried out by formulating by-laws under the "polluter-pays-principle" for private corporations or companies and business owners to partake in improving the waste management mechanism as part of their social responsibility. This will optimally reduce the trade-offs of any non-essential waste generation in commercial industry and household purchases. Meanwhile, pro-environmental product design guidelines should be formulated by environmental agencies and endorsed by local authorities for product manufacturers to comply with.

The current annual report of the Manjung Municipal Council has revealed no information regarding the expenditure of the service delivered. It is crucial for the local authority to share their expenditure (to be transparent throughout the administration) via effective means such as Gender Responsive and Participatory Budgeting (GRPB). GRPB has been practised by Penang's state and local governments with the collaboration of Penang Women's Development Corporation as an effective tool in making the budget gender responsive prudent and, most importantly, sustainable [62]. As this type of activity is voluntary-based, households' participation can determine how their tax can be used to create other public benefits from the budget they have saved through R&SS implementation, improving their willingness to be involved.

Small actions are negligible and simple but vital to shape good behavioural change in the long run. These small actions include everyday actions such as putting reusable

containers or shopping bags in the usual hand carry or the vehicle itself, saying "no" to plastics, and more. Furthermore, they require constant reminders to eventually develop a habit over time. This can help those who experience difficulties in changing and adapting to a huge change in their waste management pattern. To amplify this effect, the surrounding environment should also be designed to influence consumers, such as designing signage at the doorstep to remind oneself to bring reusables before leaving the house when takeaway is planned. Essential destinations such as schools and the workplace, where people spend a significant number of hours in, should also be redesigned. Small tips or life hacks are essential to foster creative thinking upon negligible things or happenings in life to ensure these small actions become a routine and habit [63]. This strategy can be widely promoted using both online and offline platforms (considering not all households have access to online networks).

The survey findings reveal that most of the respondents acknowledged the importance of environmental syllabus in schools and universities. These educational facilities should raise the impact of their education and research by providing the know-how for effective implementation [64]. The knowledge transfer and skill training of formal education has a great influence among youth who are more flexible towards changes in their lifestyle [48,65]. The attitude is more effectively shaped at a younger age and recognition should be given (preferably by the government or relevant industrial players) to reward their effort and strengthen their intention to commit to R&SS practices. Besides classroom teaching and social activities, competitions (public speaking, essay writing, art- and craft-related events, photography, or videography) should be held to boost the capacity, as well as interpersonal skills of the students at all ages, especially sixth formers (pre-university) and undergraduates. Placing young children in a more competitive ground can also strengthen their intentions to advocate for the cause they believe in. Youth ambassadors in promoting R&SS practices can be one of the latest trends to promote households' participation, as young leaders often appear more inspiring and exhibit greater charisma to foster social innovations. Nevertheless, the quality of persuasion should be governed to ensure quality information processing.

It will be difficult for these recommendations to achieve optimum results if the external factors are not taken into consideration. An effective support system throughout waste management stages must be built to provide a convenient environment for households to conduct R&SS by increasing the accessibility to such services. Households' waste storage system should expand along with the waste collection system. By indicating a specific day for a specific waste category collection, households can save time for waste transportation to recycling vendors, maximising ease and convenience. With the effective feedback loop system, households can sort waste correctly, while weight can be recorded into the same system (mobile application) for further network and database establishment.

During the pandemic, the new normal indeed reshaped people's lifestyles, especially considering that digital transformation has influenced how people eat, shop, and pay bills. Manjung Municipal Council has developed a mobile application, myMANJUNG, for tax collection and bill payment [66]. This intervention can be included into the system for constructive management (e.g., ability to identify ownership of sorted waste and accountability). Nevertheless, this improvement requires consistent monitoring and evaluation by trained human resources, whether through a law enforcer hired by the municipality or collaboration with NGOs (NPOs) or community associations. Tangible incentives such as vouchers or cash rebates (based on the total weight of the sorted waste collected) can reinforce households' actions by eliminating hesitancy in visualising R&SS benefits (economic). Taiwan implements an integrated waste management system, which is designed to provide households with great support to recycle and separate waste conveniently. Frequent waste collection and households can track the service trucks in real time through mobile application. These service trucks accept wide waste categories, including biodegradable waste. Designated garbage bags are used to implement a Pay as You Throw (PAYT) scheme to incorporate the responsibility of households in waste reduction [67]. This integrated

waste management system can effortlessly maximise the possibility of R&SS integration and transition from non-sanitary landfill to sustainable waste management.

Finally, the throw-away culture should end through the enforcement of law, an example from Shanghai mandatory waste separation in 2019. However, due to political complications, the local authority can instead implement a by-law to restrict non-essential product consumption. For instance, Penang has increased the price of a plastic bag from RM0.20 to RM1.00, suggesting that people can no longer take this convenience for granted [68]. A higher cost will be incurred for more damage inflicted towards the environment (more waste generated). This is necessary for households to abstain from using unsustainable products and ultimately revise their purchasing decisions. The ideal situation is the total ban of non-essential products (e.g., single-use and disposable plastic) by shifting focus and business models to the circular economy [3,6,9]; at the same time, society should be prepared with adequate knowledge and a convenient environment in which to practice R&SS with continuous motivation or feelings of commitment [33,39].

## 5. Conclusions

There was a gap observed between theory and practice in this study where households' level of knowledge, attitude and behaviour had no linear associations when tested with Pearson's Correlation Coefficient. Moderate knowledge with a positive attitude were insufficient to drive good behaviours different than that in theory. Therefore, the paper further examined the potential parameters that would shape households' behaviour and thus encourage behavioural change, considering the fact that about two-fifth of recyclers adopted source separation despite the current less-supported waste management mechanisms and environment. It was revealed that environmental protection was the most important enabling factor for households' participation in R&SS, while on the contrary, time consumption and accessibility to R&SS services were among the highest rated barriers by the non-recyclers or non-waste sorters.

There was also disconnection observed between households and the local authority where the former was unable to identify existing municipal R&SS initiatives, while the latter faced obstacles to find effective ways to engage with more diverse household groups regardless of cultural and social background. The local authority also pointed out the important roles played by the NGO in bridging the communication and trust gaps between the community and government.

This paper aimed at providing recommendations to the specific stakeholders in Manjung district to improve waste management through R&SS implementation by analysing the causations between KAB at a household level. By addressing the gap between households' awareness, more effective and targeted strategies and initiatives can be formulated to tackle the actual gap. Nonetheless, with the limited number of samplings as identified in the methodology section, further study is needed to capture more data, which can potentially be expandable to cover other districts in Perak state, or in other states in the country. In addition, as identified in this study, further study on the influence of both internal and external factors will provide in-depth understanding of the linkage between KAB in any study area.

**Author Contributions:** The paper is based on the undergraduate final year academic project of P.L.Y.; writing—original draft, H.C.G., P.L.Y.; writing—review and editing, P.L.Y., N.A.G., M.A. and H.C.G. All authors have read and agreed to the published version of the manuscript.

**Funding:** This research received no external funding.

**Institutional Review Board Statement:** Not applicable.

**Informed Consent Statement:** Not applicable.

**Data Availability Statement:** Not applicable.

**Acknowledgments:** The paper was drafted by the corresponding author during her writing residency with the company of Abu Bijan at Rimbun Dahan.

**Conflicts of Interest:** The authors declare no conflict of interest.

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
