# Peer review of "Understanding the Human Dimensions of Recycling and Source Separation Practices at the Household Level: An Evidence in Perak, Malaysia"

_sustainability, doi:10.3390/su14138023_

Round 1

Reviewer 1 Report

Dear authors

It is pleasure to review your manuscript. I have some comments to improve it.

1. The title must include the country.

2. The keywords must include the country.

3. Please detail the validation process of the 3 instruments of collection of data.

4. In the introduction it is important to include studies that can show the problem of waste and the importance of the circular economy. We suggest the next documents add in the manuscript:
https://doi.org/10.1016/j.polymdegradstab.2019.05.014
https://doi.org/10.1007/978-981-19-0549-0_9
https://doi.org/10.1007/978-981-16-4230-2_23  

5. I consider that the discussion must include more references (the whole document only has 35).

Author Response

"Please see the attachment (Cover Letter and Authors' Response to Reviewer 1's Comments."

Reviewer 2 Report

The research question is topical and brings a novel approach regarding the impact of Knowledge management, attitude management and behavior management in real life.

The scope of the research was to identify the gap between theory and practice and necessity regarding the “human dimension towards waste generation and management”.

The Title –which has a different format – should be reformulated in order to express better the focus of the paper. Maybe, this formulation should help build a more attractive title: “human dimension towards waste generation and management”.

Abstract: Even if it is brief, the abstract has a standard formulation and presents concisely the purpose and the results of the research. A brief note about the methodology should be included in the abstract. There is too long a description of the topic.

KEYWORDS – are accordingly.

The Introduction: is complying to its scope and insists on convincing about the importance that the population needs to learn more about the actual sustainable waste disposal.  

The Introduction – includes also a “Literature review”.

The introduction presents also the Research Questions – which are not appropriate. The Scope of the research should be stated at the most.

However, RQ no 2 – should be split into 2 separate questions to deal with 2 opposite concepts: “(2) What are the barriers and enabling factors that contribute to the households’ behaviour on the R&SS practices? ”

The literature review (from the Introduction section) should be improved with literature in the field of the concepts: of knowledge management, attitude management, and recycling behavior management.

“The university role in developing the human capital for a sustainable bioeconomy”

Literature Revies – Context

The literature review should be enriched with references for supporting the same concept or idea/ or theory. As we mentioned, it is included in the Introduction section.

Materials and methods / Research design

This section is atypically structured and presented.

Research methodology is presented in detail, and clearly enough in order to understand the research approach and implementation.

Results

The Results section is highly well presented in paragraphs related to each investigated component and according to each research question.

The presentation of the results is very well detailed and argued.

Discussions

This section is oriented towards the presentation of qualitative results and interview comments. Maybe these responses should be summarized into a structured table of responses or points of view.

Within this section, the authors present a large number of recommendations for the population and local authorities in order to improve the practice of R&SS.

Conclusions

Conclusions are briefly formulated but it is ok in the context of the previous sections.

References

References are rather centered on a certain area of the globe and they should reflect wider and more diverse views. Could be improved.

Author Response

"Please see the attachment (Cover Letter and Authors' Response to Reviewer 2's comments."

Reviewer 3 Report

Thank you for letting me review this very interesting manuscript, that has the potential to make a unique contribution to the field. However, I have recommended ‘major revisions’ as I feel the paper needs to be much improved to reach its full potential. See below for more detailed comments.

The Introduction contains many value laden statements with normative implications. When describing why people engage in recycling. All responsibility is placed on the induvial and not the system that allows or even promotes this behavior. I believe this is far to normative and fails to understand the complexities between attitudes, knowledge and behavior. It is stated that knowledge is important, but there are studies that indicate that knowledge is not the main driver of pro-environmental behavior. One is this one: Vicente-Molina, M. A., Fernández-Sáinz, A., & Izagirre-Olaizola, J. (2013). Environmental knowledge and other variables affecting pro-environmental behaviour: comparison of university students from emerging and advanced countries. Journal of Cleaner Production, 61, 130-138and here’s another: Liu, P., Teng, M., & Han, C. (2020). How does environmental knowledge translate into pro-environmental behaviors?: The mediating role of environmental attitudes and behavioral intentions. Science of the total environment, 728, 138126. Knowledge is not uncomplicated and is mediated by other factors. Much has been written on this area, so I urge the authors to have another look at the literature, where Shove, E. (2010). Beyond the ABC: climate change policy and theories of social change. Environment and planning A, 42(6), 1273-1285, can be a good start.

As the authors identify that the problem of proper landfills is a national issue, also lacking national processes for recycling waste not fair to place the responsibility on individual consumers. My hunch would be that if processes are failing, the lack of engagement might be a reflection on the low trust placed in authorities, not over consumption.

An article/study published in The Guardian is referred to, but this is not a scientific reference. Please find original source or remove it.

Data and analysis

I understand and respect the aim to triangulate data with multiple data points: surveys, interviews and observations. However, there are strict methodological considerations for all of these and I do not see them reported. For example; the estimated number of respondents is about 350, but only 100 answered. However, in the tables, the “N” varies from 59 to 114 – what is the reason for this?

Out of these, seven participated in interviews. In addition, it is admirable that the authors wanted to include representatives from local governance and relevant businesses, but only two responded. So how many interviews were conducted in total, what was the duration and what questions were asked? Observation data is not described well enough and there are protocols to be followed when conducting observational studies, but these are not mentioned?

I also have some concerns about the phrasing of the response alternatives in table 6, but his might be a translation issue. The response categories have the phrasing “important”, but the items are not formulated in a way that would make this a logical response (e.g. the first item is “I am responsible for the waste that I generate”). In addition, if some scales are five-point Likert scale, why use correlation and not analysis of variance? Also, the waste collection items are not very consistent. The first measures willingness to pay, the second level of satisfaction with the sorting system. These are fine as stand-alone questions btu measures different aspects that cannot (in my view) be grouped together. This makes me wonder about the reliability of the study.

The Results are very detailed, which is good, but tat the same time somewhat confusing. For example, what is the difference between Frequency and Response (in table 16 for example)? I cannot find any explanation in the text – but perhaps I’m just not seeing it. Please provide a better explanation for how results are presented.  Using only frequencies can be fine, but I fail to see how the results presented and analysis used links to the original research question.

Conclusions need to be better linked to the results. For example, in the results we can see that 13% think that waste is handled well, whereas almost 80% feel that is it important for the environment. I would think that this is much more interesting that factual knowledge about the recycling itself. If there is low trust among the residents in that the municipality is handling waste well, this might play a larger part in influencing the behavior than factual knowledge.

In addition, in the Discussion, the authors introduce us to what the respondents stated in the interview. I would have liked to see that in the Results, in a separate sub-heading.

In sum, I believe this paper has a lot of interesting results, but that the paper needs to be restructured to do that justice. The Introduction fails to mention some of the relevant factors that are then measures in the study. One way of improving it, in my opinion, is to have a more open approach to the research question: since studies for this part of the world are scarce, it might suffice to keep the introduction more general (and not only focusing on knowledge) but include other factors as well: attitudes, perceptions of responsibility and trust in authorities and /or waste management practices. The Method section needs to give us a better understanding of what measurements and data points were used.  The Results needs better presentation and clarity, as well as a more thorough statistical discussion. The data is very rich and should be given more justice. Also, interview data should be presented here. If the authors chose to follow my suggestion about widening the Introduction and scope of the article, the Discussion can be structured to illustrate factors that influence and factors that deter recycling and link it to literature that provides explanations and suggestion on ways forward. I encourage the authors to rework their contribution as I feel they have valuable data.

Author Response

"Please see the attachment (Cover Letter and Authors' Response to Reviewer 3's comments."

Round 2

Reviewer 1 Report

I recommend the approval of the article.

Reviewer 3 Report

Se attached file
